# Health Disparities in Multiple Sclerosis among Hispanic and Black Populations in the United States

**DOI:** 10.3390/biomedicines11041227

**Published:** 2023-04-20

**Authors:** Michael Z. Moore, Carlos A. Pérez, George J. Hutton, Hemali Patel, Fernando X. Cuascut

**Affiliations:** Maxine Mesinger Multiple Sclerosis Center, Baylor College of Medicine, Houston, TX 77030, USA

**Keywords:** MS—multiple sclerosis, DMT—disease modifying therapies, pwMS—persons with multiple sclerosis, SDOH—social determinants of health, EDSS—Expanded Disability Severity Scale, SES—socioeconomic status, nSES—neighborhood socioeconomic status

## Abstract

Multiple sclerosis (MS) is an acquired demyelinating disease of the central nervous system (CNS). Historically, research on MS has focused on White persons with MS. This preponderance of representation has important possible implications for minority populations with MS, from developing effective therapeutic agents to understanding the role of unique constellations of social determinants of health. A growing body of literature involving persons of historically underrepresented races and ethnicities in the field of multiple sclerosis is assembling. Our purpose in this narrative review is to highlight two populations in the United States: Black and Hispanic persons with multiple sclerosis. We will review the current understanding about the patterns of disease presentation, genetic considerations, response to treatment, roles of social determinants of health, and healthcare utilization. In addition, we explore future directions of inquiry as well as practical methods of meeting these challenges.

## 1. Introduction

Multiple sclerosis (MS) is an acquired demyelinating disease of the central nervous system (CNS). A recent study estimates that in 2017, nearly 1 million adults had MS in the United States (US) [1]. The burden of this disease is well established, as it represents the most common progressive neurologic disorder in young adults worldwide [1]. The most recent estimated total economic burden in the United States (US) exceeded $85 billion [2]. However, there has been great pharmacologic advancement in the treatment of MS over the past few decades, and MS is now considered a less disabling condition with appropriate intervention [3].

Previous studies have shown that different genetic, environmental, and immunological risk factors play a role in the pathogenesis of the disease. Moreover, MS may be influenced by multiple social determinants of health, such as income, education, and cultural perceptions [4,5,6,7]. These have ranged from decreased access to healthcare, increased frequency of lower incomes correlating with increased disease severity, and decreased representation in clinical trials.

Increasing evidence suggests that racial disparities are important factors that may explain differences in the disease course, incidence, prevalence, and outcomes [8]. Still, Black and Hispanic populations in the United States remain largely underrepresented and understudied in clinical trials [9,10,11,12]. Recently, a cumulative number of studies have been focusing on improving our understanding of the disease course and disparities across underrepresented populations [13]. A more comprehensive understanding of the disparities can help improve disease management in patients of diverse backgrounds. This narrative review aims to explore these topics as it applies to historically understudied minority populations in the US, specifically Black and Hispanic persons with multiple sclerosis (pwMS).

## 2. Methods/Definitions

Utilizing the Texas Medical Center Library OneSearch and Pubmed, we performed a literature search for English language studies including the phrases “multiple sclerosis”, “disparities”, “social determinants of health”, “African-American”, “Black”, “Hispanic”, “Latino”, “LatinX” from 2014 to the date of search in October 2022, and similar searches were carried out periodically in subsequent reviews of the text. Additional studies were reviewed guided by the reference lists of relevant studies and further elements of interest emerging from this literature review.

We will use Black, Hispanic, and White racial/ethnic designations for this paper. To ensure accuracy, the term Black is substituted to reference patients defined as “African-American” or “Non-Hispanic Black,” White is substituted for “Non-Hispanic White,” and Hispanic for those of Latin American descent. While the term also applies to persons from Spain, we will limit its use to the Latin American population in this article. If a population in a study does not meet one of these designations, or if the use of these general terms would impair the appropriate interpretation of the study in question, we will revert to the terminology used by the original authors. While a diverse array of populations of different races are affected by MS and important considerations likely exist in the management of MS, the focus in this review is largely limited to the three populations detailed above under their positions as those with the highest representation of MS disease burden in the United States.

## 3. Multiple Sclerosis & Its Course in Black Persons

Historically, the incidence of MS in the US was thought to be substantially higher in White persons of European background than in most minority groups, including Black individuals. More recent studies have challenged this position. A large cohort study from 1990 to 2007 within the United States military served by the Veterans Affairs system demonstrated an incidence rate of 12.1 per 100,000 among Black persons, compared to an incidence of 9.3 per 100,000 among White persons [14]. This is further corroborated by a retrospective cohort study in Southern California that reported an incidence of MS in Blacks that exceeds that of Whites (10.2 per 100,000 and 6.9 per 100,000, respectively) [15].

In addition to the increased incidence rate, Black pwMS may have a less favorable disease course and morbidity than their White counterparts. For example, a 2003 study among patients of the New York State Multiple Sclerosis Consortium (NYSMSC) demonstrated a younger age at diagnosis and greater disability with increased disease duration among Black pwMS [16]. Other studies have corroborated these data and have demonstrated faster progression to require ambulatory assistance [17,18]. Furthermore, other studies showed that Blacks consistently had higher disability than Whites when measuring cognitive, ambulatory, and manual dexterity functions [19].

Black pwMS have been noted to be more likely to have spinal cord lesions and multifocal clinical presentations [17]. They also were less likely to experience full recovery from relapses, had shorter times to a second relapse, and had more frequent relapses overall [16]. A faster transition from relapsing–remitting MS (RRMS) to secondary progressive MS (SPMS) was also observed in Black pwMS [20]. Moreover, a multicenter study across 3 academic hospitals revealed significantly lower visual acuity in Black patients in both high- and low-contrast visual acuity, as well as faster retinal nerve fiber layer (RNFL) thinning [21].

MRI volumetrics and lesion burden studies have also suggested less favorable outcomes for Black pwMS. One study demonstrated that Black pwMS had diffusely lower brain volumes, more brain lesions on MRI, lower gray matter volumes, and lower cortical and thalamic volumes [19]. An additional study comparing MRIs between Black and White pwMS demonstrated a higher burden of T2 hyperintense lesions and T1 hypointense lesions in the former group, which correlate with greater associated disability [22].

## 4. Multiple Sclerosis & Its Course in Hispanic Persons

Available literature on MS and its course in Hispanic pwMS is more limited than in White and Black pwMS in the US. Historically, data have largely been drawn from patient registries and retrospective cohort studies. As previously stated, there is an additional limitation in identifying consistent patient grouping due to variations in terminology (Hispanic, Latino, LatinX, etc.). With that qualification, available population studies seem to reflect a lower incidence and prevalence in Hispanic populations than in White and Black Americans [14,15,23,24,25]. Similarly, mortality associated with MS, as measured by death certificate analysis, appears lower in Hispanic pwMS compared to their Black and White counterparts [8].

Over the last few decades, cohort study observations regarding morbidity, disease course, and severity across racial groups have shown some inconsistencies. A study utilizing retrospective chart review and patient interviews in southern California demonstrated earlier age of disease onset and diagnosis and more frequent myelitis presentations among Hispanics. Hispanic and White pwMS appeared to have similar rates of ambulatory disability [26]. In contrast, a study comparing disability as measured by the Patient Derived Multiple Sclerosis Severity Score (PDMSS) demonstrated higher disability in Hispanic pwMS compared to White pwMS, following a trajectory similar to Black pwMS [27]. In a single institution cross-sectional study involving primarily Hispanic pwMS of Caribbean descent, an increase in ambulatory disability was reported in Hispanics compared to White pwMS [28]. A separate single academic center registry review found that, like Blacks, Hispanic pwMS were more likely to have a lower survival time ratio than Whites [29]. They were also more likely to present at a younger age and to have optic neuritis at presentation [29]. Additionally, Hispanic pwMS have been found to have significantly lower baseline thalamic volume measures correlating with a higher median baseline Expanded Disability Severity Scale (EDSS) [30]. Taken together, these studies suggest differences in the type and degree of morbidity associated with MS in Hispanic populations.

## 5. Genetics

While the immunopathophysiology and phenotypic expression of MS is complex, genetic factors play a substantial role. Variations in the human leukocyte antigen (HLA) complex genotypes have been shown to correlate with differences in the incidence of MS [31]. For example, HLA DRB1*15:01 is associated with an increased risk of disease development, while HLA-A*02 is associated with a decreased corresponding risk [5]. Extending beyond the HLA complex, single nucleotide polymorphisms (SNPs) in other exome regions have been associated with less significant effects on the risk of MS development [7].

It should be recognized that most of these studies have been carried out in persons of European ancestry, and it should not be taken for granted that these genetic risk factors are equally present in or have an equivalent effect on those of different races and ethnicities.

Recent studies have utilized genotyping within a broader range of races, whole exome sequencing, and tools such as admixture mapping to correlate disease incidence and expression with trends in ancestral haplotypes in key regions. In a study involving HLA genotyping and admixture mapping in Black and White pwMS, HLA DRB1*15 was appreciated in both groups. It was present at a higher frequency in pwMS compared to the unaffected controls [32]. In this study, admixture evidence of African origin at HLA DRB1*15 was associated with higher disability as measured by Multiple Sclerosis Severity Score (MSSS) after controlling for other factors. However, more recent studies using admixture mapping have demonstrated the opposite correlation with incidence. In an admixture mapping study involving Black, Hispanic, and Asian pwMS in the US, individuals with the European haplotype of HLA DRB1*15 experienced three times the disease risk compared to those with the African haplotype [33]. Similar literature on specific genetic variation in the Hispanic pwMS is somewhat more limited.

Outside the HLA complex, ancestral variation in other areas is also being studied. A whole exome scan of a sample of Black pwMS in the US indicates an increased risk for MS development in those who inherit European haplotypes in two loci on chromosome 1 [34]. Within a previously mentioned study, a genome-wide admixture search demonstrated a new region of interest on chromosome 8 with a significantly higher representation of European origin haplotypes in Hispanic pwMS than in controls [33]. A different study utilizing a custom SNP array to investigate a sample set of Black pwMS (2319 total, 803 with MS, 1516 without) found significant overlap in the variants associated with increased risk in SNPs previously identified in those of primarily European ancestry [35]. These findings were not corroborated by a more recent study in Black, Hispanic, and Asian pwMS and controls in the US, in which the majority of the 200 SNPs of interest identified in those of European ancestry were not significantly associated with MS [33].

Multiple considerations above invite further inquiry, including the prospect of genetic variations specific to minority populations and mixed evidence in admixture mapping and SNP data. Further progress in this area could have future implications in pharmacogenomics and personalized medicine in MS.

## 6. Response to Disease Modifying Therapies

Much of our understanding of MS clinical course in minority populations comes from limited studies. A systematic review of phase III disease-modifying treatment (DMT) clinical trials found that most trial subjects were White, raising questions about expectations of treatment tolerability and effectiveness across minorities [36]. These questions are rendered more compelling by evidence from post hoc drug trial analyses demonstrating increased rates of disease progression and lesion formation in Black pwMS treated with interferon-β-1a [37,38]. While further investigations into other DMTs have commenced, they have generally consisted of post hoc analyses. These have demonstrated similar treatment benefits in the case of natalizumab and ocrelizumab in trials with Black participants, though with slightly more adverse events and hypersensitivity reactions in the case of ocrelizumab [38,39]. In the case of dimethyl fumarate, post hoc analyses of phase III clinical studies appeared to demonstrate similar clinical efficacy and tolerability among Black, Hispanic, and Asian pwMS subgroups as appreciated in the full study population [40]. Similarly, subgroup analyses of a multinational prospective cohort study on pwMS treated with dimethyl fumarate found similar relapse rate reductions and safety profiles in Hispanic and Black pwMS compared to non-Hispanic and non-Black pwMS. However, they demonstrated a less pronounced lymphocyte count decrease in Blacks [41,42].

It remains difficult to cite discrete mechanisms to account for differences in disease severity and response to DMTs among these populations and literature on this topic remains sparse. However, there is some evidence of factors that could underlie these disparities. For example, a cross-sectional study of White, Black, and Hispanic pwMS off DMT or on natalizumab compared to healthy controls of matched self-reported racial/ethnic background demonstrated a significant elevation in circulating plasmablasts among Black and Hispanic pwMS compared to White pwMS [11].

Observational studies not focused on a single DMT have also raised concerns about worsened outcomes. In a single academic center retrospective electronic medical record review (EMR) comparing sociodemographic characteristics, treatment response, and disability outcomes between 300 age and gender-matched cohorts, Hispanic and Black pwMS had a higher median EDSS score at baseline. This was despite similar patterns of DMTs used, average therapeutic lag, duration of individual therapies, and frequency of medication change [29]. Altogether, these studies underscore the importance of further research on the effectiveness and tolerability of existing DMTs in non-White populations and increased minority representation in future DMT trials [13].

## 7. Social Determinants of Health and Healthcare Utilization

There is increasing awareness about the impact of social determinants of health (SDOH), such as where a person is born, grows, works, lives, and ages, amidst the wider set of forces and systems shaping the conditions of daily life, on disease outcome and quality of life in MS [43,44]. Though the impact of these factors remains incompletely understood, available data suggest that significant differences in SDOH exist across racial groups [4]. Differential patterns and degree of healthcare utilization across minority populations have been described [45]. However, current data regarding these trends and their effects on minority populations among pwMS are limited.

Regarding general neurologic care, significant disparities in healthcare utilization between Whites and minority populations have been described. Data from the Medical Expenditure Panel Survey (MEPS) from 2006 to 2013 involving patients with neurologic conditions demonstrated that Blacks and Hispanics were approximately 30% and 40% less likely to see an outpatient neurologist compared to Whites, respectively [45]. This disparity has also been reported in pwMS, as demonstrated in a national cross-sectional study showing that the probability of seeing a neurologist was significantly lower among patients who did not have health insurance, were poor, lived in rural areas, and, notably, Black individuals [46]. Since adequate MS treatment requires a comprehensive and multidisciplinary approach, the presence of these risk factors can suggest a potential increased risk of disability and disease progression in minority populations. In a previous single-center retrospective study, after adjustment for race and age, MS patients who were evaluated by a neurologist at diagnosis were found to have 60% lower odds of attaining an EDSS > 4.5 at subsequent follow-up visits compared to patients evaluated by a non-neurologist [47].

Nevertheless, disparities in healthcare utilization appear to extend beyond direct access to care by a neurologist into other areas equally worth noting. In this regard, community-based services, in a study of Medicare/Medicaid eligible pwMS who were identified as Black or White (race limited to a dichotomous variable), Black pwMS utilized multiple services less frequently than their White counterparts, including case management (64% less likely), nursing services (48% less likely), and equipment, technology, and modification services (31% less likely) [48]. This trend mirrors previously reported trends toward greater disability among Black pwMS. In fact, in this study, Black pwMS utilizing Medicaid-funded services were more likely to have mobility impairment despite a younger average age than their White counterparts [48].

Important correlations have emerged regarding income and its effect on morbidity and management. White patients with lower socioeconomic status (SES) had a worse mean performance score on walking, manual dexterity, and cognitive processing tests [19]. One study of approximately 8500 individuals showed that Black pwMS were more likely to have Medicaid, be unemployed, and have higher rates of disability, both on self-reported and objective measures [19].

The area deprivation index (ADI) is a widely used and multidimensional tool comprised of seventeen variables to assess for socioeconomic disadvantage [49]. In White pwMS, higher ADI scores (indicating higher disadvantage) correlated with slower cognitive processing and manual dexterity speeds. A correlation was also noted between slower walking speeds and residence in low-income households. In Black participants, lower household income was associated with slower manual dexterity, but there was no clear association between ADI scores and other neurologic performance measures [19]. Similarly, correlations between income and mental health comorbidities in pwMS have been described. A study involving pwMS identified as Black, Hispanic, or White correlated neighborhood socioeconomic status (nSES) scores with results of screening tests for multiple mental health disorders, including anxiety, depression, and alcohol use disorder. This study found that patients in the lowest quartile of nSES (lower relative income) in all groups had lower screening test scores for mental health disorders overall [50]. In addition, Black and Hispanic pwMS more frequently occupied the lower quartile of nSES in this study population.

Of particular interest is the role of perception of healthcare and MS research participation among Black and Hispanic pwMS. Specifically, in a web-based survey among members of the MS Minority Research Engagement Partnership Network (MSREPN), important differences between Hispanic, Black, and White participants were described. Specifically, Hispanic and Black participants more frequently reported concerns of “being taken advantage of or used by” the research team than White participants. Hispanic participants more frequently reported concern about research participation affecting their legal status [51].

Additional social determinants of health, including cultural dispositions towards illness and disease mitigation, health literacy, regional/geographical barriers to care, insurance coverage, and support community involvement have been identified [4]. Increased research into the type and degree of effect of these factors on disease management among historically underrepresented populations in MS care is necessary. Healthcare systems serving these populations should be structured to account for these differences.

## 8. Limitations

There are several limitations to this narrative review. First is the limitation of the focus to a particular region (i.e., United States) and the limited subset of races and ethnicities (i.e., White, Black, Hispanic). It is important to recognize that many races are not reviewed here, many of whom have a presence in the United States and have unique trends in disease incidence, phenotype, morbidity, genetic/epigenetic factors, and social determinants. Their omission from this review is due in part to a present paucity of data relative to the populations discussed herein, highlighting the importance of further and more inclusive studies in this field. Additionally, in the formation of this narrative review, there is the prospect of sampling bias of included articles.

Importantly, in a review of the literature on historically understudied racial groups, the use of inconsistent terminology used to describe different populations presents itself. Though race identification is a heavily personal decision, issues in the terms used can complicate the interpretation of results. Terminology referring to minority populations in the US has been inconsistent. Various terms are used to refer to different populations in the literature, including African-American, Afro-Caribbean, Black, White, Caucasian, Hispanic, Hispanic Black, Hispanic White, Hispanic non-White, Mestizo, Latino, LatinX, etc. We must practice caution when analyzing these terms as they are not synonymous. For example, as a matter of etymological integrity, the term Hispanic is not used to refer to people from Brazil, as the term carries a linguistic association with Spanish-speaking countries. The term African-American usually pertains to those with lineage tracing to Africa, but it might not apply to those whose known lineage traces no further than the country in which they live. Another important distinction is that some studies classify individuals by self-identified methodology, whereas in other studies authors assign these classifications. Even in the cases of self-identification, there exists a variance in the options provided and their range and clarity. This presents a difficulty in properly representing, comparing, and contrasting population trends between studies over different regions and periods. In attempting to identify trends and patterns, we risk false equivalencies as a function of unclear terminology about race and ethnicity.

## 9. Discussion and Future Directions

Emerging literature demonstrates important differences in incidence, prevalence, phenotypic features, and associated disability in Hispanic and Black pwMS as compared to White pwMS as well as unique elements in genetic predisposing features and risk factors across these populations. Moreover, an increasing number of community-based MS observational studies describe important racial and ethnic disparities in SDOH.

Considering these differences, we would like to identify future directions of research and practice strategies to better serve these populations (Figure 1). First, increased efforts are needed in education amongst healthcare professionals regarding the incidence and prevalence of MS in minority populations. This may decrease unconscious diagnostic biases regarding pretest probability delaying timely diagnosis [52]. This is particularly important among Black pwMS given estimates of incidence now exceeding those of their White counterparts. Additionally, healthcare providers’ awareness of the variability in disease severity and expression among Black and Hispanic pwMS may improve proper comprehensive care and patient counseling.

Next, increased enrollment of minority pwMS in registries can assist in advancing research efforts among minority populations. The evidence of variation in response to some DMTs among minority groups underscores the importance of increased representation of non-White pwMS in clinical trials. There is evidence of recognition of this need and attempts to better meet it in the design of more recent drug trials, including those involving multiple sclerosis therapeutics [12]. To this end, investigators and participating clinical study sites should be conscious of differences in the perception of healthcare research by study participants. These can be addressed in several ways: increased representation of these populations amongst investigators and associated healthcare teams, outreach in minority communities and collaboration with community leaders, and increased availability of counseling and educational materials appropriate to the participant’s native language and degree of health literacy. With these and other measures in place, increased minority representation in DMT clinical trials and future research in genetic/epigenetic variation across non-White pwMS can help advance pharmacogenomics among historically underserved populations.

Considering the differences in SDOH and healthcare utilization patterns in Black and Hispanic pwMS, many practical adjustments in care strategies may prove useful. A few publications have outlined some measures in improving care with a view to these needs, but this remains a poorly understood topic [4,13]. Reliable communication strategies should be established between healthcare providers and patients to promote comprehensive outpatient care and minimize unnecessary hospital visits and economic burden. Ideally, pwMS should have an associated healthcare provider such as a nurse navigator to decrease this reliance further. Likewise, there is evidence that co-locating allied care services (mental health, physical medicine and rehabilitation, etc.) may increase utilization among Black and Hispanic pwMS [50]. In a previously mentioned report on correlations of income and mental health comorbidities, Black and Hispanic pwMS were more likely to seek mental health care if it was co-located with sites at which their MS care was provided [50].

In addition to the elements described above, practical measures should be placed to best meet the needs of minority pwMS. Strategies to develop cultural competence among care teams should be employed to detect and correct possible deficiencies. Proactive outreach in communities with high minority representation by MS societies and centers of excellence may have many benefits, including increasing community awareness and fundraising efforts. Attention should also be paid to decreasing obstacles in access to care. Potential strategies include increased utilization of telemedicine, satellite clinic sites, and cultivating practical referral networks from primary care providers.

## 10. Conclusions

There is increasing recognition of significant differences in incidence patterns, prevalence, and outcomes in Black and Hispanic pwMS. However, the underpinnings of these differences remain an area that requires future study. The role of genetic and biological determinants of health has long been understudied and therefore underrecognized. Race and ethnicity are also associated with genetic ancestry and therefore indirectly related to genetic variants that may affect disease and health outcomes in MS. Discrepancies in socioeconomic constructs are becoming important interventional targets to help alleviate disease severity by race and ethnicity. While further research remains a high priority in this field, evidence now exists to guide improvements in the delivery of MS care to historically underserved minority populations.

## Figures and Tables

**Figure 1 biomedicines-11-01227-f001:**
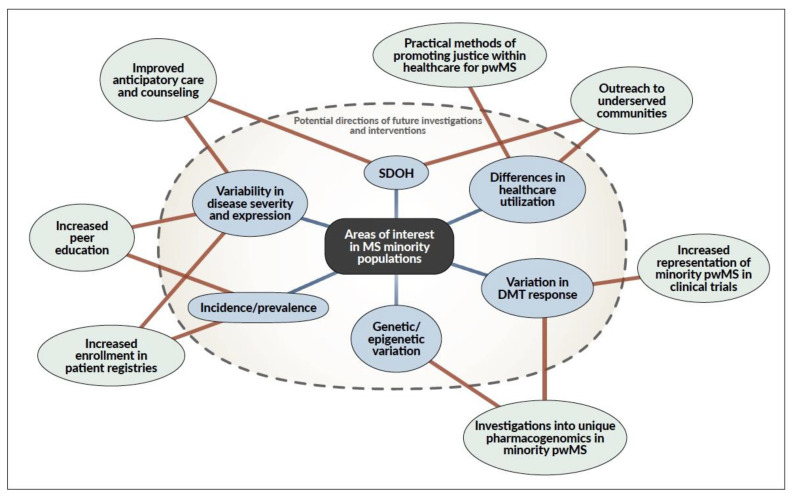
Areas of emphasis in care of minority populations with multiple sclerosis and future directions. The diagram illustrates points of emphasis emerging from a review of literature on multiple sclerosis (MS) in minority populations, specifically Hispanic and Black persons with multiple sclerosis (pwMS). Items beyond the dashed line indicate areas of future research and practice strategy changes.

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
