# Peer review of "Health Disparities in Multiple Sclerosis among Hispanic and Black Populations in the United States"

_biomedicines, 2023, doi:10.3390/biomedicines11041227_

Round 1

Reviewer 1 Report

Ιnteresting article about the differences between MS patients of different nationalities. The design of the review needs to be reformed with tables clearly showing the parameters tested, the differences between whites and other ethnicities and the literature source of the information. A greater body of references is needed.

Author Response

Thank you for your feedback on our paper.

Based upon your feedback, we have expanded our body of references and incorporated additional elements in the body of our review. Unfortunately, a greater expansion of the reference list is limited by the present limitation of research in this area.

Given the diversity and heterogeneity of the evidence discussed within each section this review, a narrative format was ultimately favored over summarization within a series of tables.

Many thanks for your thoughtful feedback and your contribution to an improved final product.

Reviewer 2 Report

The manuscript under analysis, “Health Disparities in Multiple Sclerosis Among Hispanic and Black Populations in the United States,” is a narrative review. Due to the very nature of the narrative review, some of its discussion elements are highlighted and presented coherently between form and content. Therefore, the text is consistent with the structure of a narrative review. Furthermore, the state of language is compatible with a narrative review, and its content developed with adequacy to the narrated phenomenon. Giving its limits and, at the same time, listing its potential for elaboration and future studies.

Congratulations to the authors. The form of discussion initiated by the form and content that the narrative review allowed was creative and sustained by its content.

My only suggestion, the authors, had already made, when they described in the discussion, the possible handling elements to improve the attention to this group of affected people. This public health problem has been increasing and is recognized as a disease that is sometimes neglected in healthcare settings worldwide.

Thank you for being able to carry out the review.

Author Response

Thank you for your feedback on our paper.

In our most recent version, we have expanded our body of references and incorporated additional elements in the body of our review. Unfortunately, a greater expansion of the reference list is limited by the present limitation of research in this area.

Many thanks for your thoughtful feedback and your contribution to an improved final product.

Round 2

Reviewer 1 Report

Accepted for publication